# Non-Surgical Bladder-Sparing Multimodal Management in Organ-Confined Urothelial Carcinoma of the Urinary Bladder: A Population-Based Analysis

**DOI:** 10.3390/cancers16071292

**Published:** 2024-03-27

**Authors:** Mario de Angelis, Andrea Baudo, Carolin Siech, Letizia Maria Ippolita Jannello, Francesco Di Bello, Jordan A. Goyal, Zhe Tian, Nicola Longo, Ottavio de Cobelli, Felix K. H. Chun, Fred Saad, Shahrokh F. Shariat, Luca Carmignani, Giorgio Gandaglia, Marco Moschini, Francesco Montorsi, Alberto Briganti, Pierre I. Karakiewicz

**Affiliations:** 1Cancer Prognostics and Health Outcomes Unit, Division of Urology, University of Montreal Health Center, Montreal, QC H2X 3E4, Canada; mariodeangelis1993@gmail.com (M.d.A.); adrea.baudo@unimi.it (A.B.); carolin.siech@kgu.de (C.S.); fran.dibello12@gmail.com (F.D.B.); jordan.goyal@umontreal.ca (J.A.G.); zhe.tian@umontreal.ca (Z.T.); fred.saad@umontreal.ca (F.S.); 2Division of Experimental Oncology, Unit of Urology, URI—Urological Research Institute, IRCCS San Raffaele Scientific Institute, 20132 Milan, Italy; gandaglia.giorgio@hsr.it (G.G.); moschini.marco@hsr.it (M.M.); montorsi-francesco@hsr.it (F.M.); briganti.alberto@hsr.it (A.B.); 3Vita-Salute San Raffaele University, 20132 Milan, Italy; 4Department of Urology, IRCCS Policlinico San Donato, 20097 Milan, Italy; luca.carmognani@unimi.it; 5Department of Urology, University Hospital, Goethe University Frankfurt, 60590 Frankfurt am Main, Germany; felix.chun@kgu.de; 6Department of Urology, IEO European Institute of Oncology, IRCCS, Via Ripamonti 435, 20141 Milan, Italy; ottavio.decobelli@ieo.it; 7Department of Neurosciences, Science of Reproduction and Odontostomatology, University of Naples Federico II, 80131 Naples, Italy; nicola.longo@unina.it; 8Department of Urology, Comprehensive Cancer Center, Medical University of Vienna, 1090 Vienna, Austria; shahrokh.shariat@meduniwien.ac.at; 9Department of Urology, Weill Cornell Medical College, New York, NY 10065, USA; 10Department of Urology, University of Texas Southwestern Medical Center, Dallas, TX 75390, USA; 11Hourani Center of Applied Scientific Research, Al-Ahliyya Amman University, Amman 19328, Jordan

**Keywords:** bladder cancer, MIBC, urothelial carcinoma, non-surgical management, trimodal therapy, TMT, chemotherapy, radiotherapy, cancer-specific mortality

## Abstract

**Simple Summary:**

Radical cystectomy represents the gold-standard treatment for organ-confined urothelial carcinoma of the urinary bladder. However, it does not represent a treatment option for some patients unfit for major surgery or wanting to preserve their bladder. In consequence, non-surgical bladder-sparing multimodal strategies are gaining popularity. Among these, the combination of transurethral resection, chemotherapy, and radiotherapy (namely trimodal therapy) represents the most recognized and validated treatment option. However, some patients may only be eligible for either chemotherapy or radiotherapy but not both. The aim of this study was to evaluate cancer-specific mortality differences among these treatments. We ascertained that when radical cystectomy is not an option, strict trimodal therapy that includes both chemotherapy and radiotherapy after transurethral resection offers the best cancer control. Additionally, when strict trimodal therapy cannot be delivered, chemotherapy represents the second-best option. Finally, radiotherapy without chemotherapy offers the worst cancer control.

**Abstract:**

Background: Trimodal therapy is considered the most validated bladder-sparing treatment in patients with organ-confined urothelial carcinoma of the urinary bladder (T2N0M0). However, scarce evidence exists regarding cancer-specific mortality (CSM) differences between trimodal therapy and other non-extirpative multimodal treatment options such as radiotherapy alone after transurethral resection (TURBT + RT) or chemotherapy alone after transurethral resection (TURBT + CT). Methods: Within the Surveillance, Epidemiology, and End Results database (2004–2020), we identified T2N0M0 patients treated with either trimodal therapy, TURBT + CT, or TURBT + RT. Temporal trends described trimodal therapy vs. TUBRT + CT vs. TURBT + RT use over time. Survival analyses consisting of Kaplan–Meier plots and multivariable Cox regression (MCR) models addressed CSM according to each treatment modality. Results: 3729 (40%) patients underwent TMT vs. 4030 (43%) TURBT + CT vs. 1599 (17%) TURBT + RT. Over time, trimodal therapy use (Estimating annual percent change, EAPC: +1.2%, *p* = 0.01) and TURBT + CT use increased (EAPC: +1.5%, *p* = 0.01). In MCR models, relative to trimodal therapy, TURBT + CT exhibited 1-14-fold higher CSM and TURBT + RT 1.68-fold higher CSM. In a subgroup analysis, TURBT + RT was associated with 1.42-fold higher CSM than TURBT + CT (*p* < 0.001). Conclusions: Strict trimodal therapy that includes both CT and RT after TURBT offers the best cancer control. When strict trimodal therapy cannot be delivered, cancer-specific survival outcomes appear to be superior with TURBT + chemotherapy compared to TURBT + RT.

## 1. Introduction

Radical cystectomy represents the gold-standard treatment for organ-confined urothelial carcinoma of the urinary bladder (namely, cT2N0M0) [1]. However, radical cystectomy is a major urologic surgery associated with complication rates ranging from 50–100% and severe complication rates (defined as Clavien-Dindo grade ≥ 3) ranging from 30–45% [2,3,4,5]. Moreover, mortality rates after radical cystectomy are recorded at between 2.1–3.2% at 30 days and 3.4–8.0% at 90 days [6,7]. As such, it may not represent a treatment option for some patients unfit for major surgery. Additionally, radical cystectomy severely affects multiple organ systems (urinary, bowel, and sexual), leading to a suboptimal quality of life after surgery, especially regarding body image and sexual function [8].

For all the above reasons, non-surgical bladder-sparing strategies are gaining popularity for patients with organ-confined urothelial carcinoma of the urinary bladder [9]. Different strategies have been investigated over the years. The principal therapeutic options in our armamentarium include transurethral resection of the bladder tumor (TURBT), chemotherapy (CT), and external beam radiation therapy (RT). However, current guidelines strongly discourage the adoption of a single therapy as a definitive treatment in patients with organ-confined urothelial carcinoma of the urinary bladder, as they are all associated with dismal cancer control outcomes [1]. For example, approximately 50% of patients treated with TURBT alone in the context of muscle-invasive bladder cancer will eventually undergo radical cystectomy during long-term follow-up, with 50% cancer-specific mortality in this group [10]. In consequence, when radical cystectomy is not an option, the combination of two or more strategies is preferred.

Multimodal bladder-sparing treatments with TURBT plus either RT or CT have previously been investigated. In a prospective Phase 2 non-randomized comparative trial, Solsona et al. evaluated long-term cancer control rates in patients treated with TURBT + CT vs. radical cystectomy. Specifically, in patients treated with TURBT + CT, five-year cancer-specific mortality and ten-year cancer-specific mortality were 64.5% and 59.8%, respectively, with no significant difference compared to the radical cystectomy arm (*p* = 0.5) [11]. Similarly, TURBT + RT has been the most used for bladder-sparing treatment in patients with organ-confined urothelial carcinoma of the urinary bladder [12]. Subsequently, several trials demonstrated the added value of chemotherapy administration in patients treated with TURBT + RT in terms of cancer-specific survival [13]. Since then, the combination of TURBT, RT, and CT, namely trimodal therapy, represented the most recognized and validated organ-sparing strategy for patients with organ-confined urothelial carcinoma of the urinary bladder [9,12,13,14,15]. The rationale of this multimodal treatment is to achieve optimal local control through the maximal resection of the bladder tumor plus the irradiation of the resection bed with or without the adjacent lymph nodes. Additionally, the inclusion of systemic chemotherapy serves a dual purpose: treating eventual micro-metastatic sites and acting as a radiosensitizer [1,9,12,13]. Despite neoadjuvant chemotherapy prior to radical cystectomy showing better cancer-control outcomes, the role and setting of systemic chemotherapy in the context of TMT are still under debate. Indeed, neoadjuvant chemotherapy with concomitant RT is associated with higher rates of locoregional control [14,16]. Conversely, chemotherapy administration was not associated with lower metastasis-free survival or disease-free survival at ten-year follow-up, thus suggesting that the role of chemotherapy should be considered more in terms of radiosensitization [14]. Nevertheless, the synergic role of all three treatment modalities resulted in five-year cancer-specific survival rates ranging from 50% to 85% and overall survival rates ranging from 35% to 75% [12,13,17,18,19].

However, of all candidates, some may tolerate RT and CT. Conversely, others may only be eligible for either CT or RT, but not both. Consequently, in patients who are candidates for a bladder-sparing multimodal treatment modality, the combination of all three is still used (TURBT + RT, TURBT + CT, and trimodal therapy).

Although multiple studies have addressed cancer-specific mortality (CSM) after trimodal therapy, none have quantified CSM according to TMT that includes concomitant RT and CT relative to CT alone or relative to RT alone, after TURBT. Moreover, no previous studies have directly compared trimodal therapy use over time relative to CT alone or RT alone after TURBT in such patients. We addressed these knowledge gaps and tested for differences in the administration of these treatments in patients with organ-confined urothelial carcinoma of the urinary bladder over time. We also hypothesized that no differences in CSM exist between TMT vs. CT or RT alone after TURBT. We relied on the most contemporary Surveillance, Epidemiology, and End Results (SEER) database (2004–2020) to test these hypotheses.

## 2. Materials and Methods

### 2.1. Population of Interest

The SEER database represents approximately 34.6% of the United States population regarding demographic characteristics and cancer incidence rates [20]. In the current manuscript, we selected patients with newly diagnosed and histologically confirmed muscle-invasive urothelial carcinoma of the urinary bladder (International Classification of Disease for Oncology [ICD-O-3] site code C67.0–C67.9) and organ-confined stage (cT2N0M0) relying on the SEER database (2004–2020). Trimodal therapy was defined as the combination of TURBT, CT, and RT. Death was defined according to the SEER mortality code as CSM (death attributable to bladder cancer) or other-cause mortality (death attributable to any other cause). The study focused on the three most prevalent non-surgical bladder-sparing multimodal treatment options for organ-confined urothelial carcinoma of the urinary bladder: trimodal therapy, TURBT + CT, and TURBT + RT. Patients who underwent partial cystectomy, radical cystectomy with variant histology, non-organ confined stage, metastatic disease, and unknown vital status information, unknown, and other treatments, as well as all autopsy or death certificate cases, were excluded. All patients needed to have undergone a pretreatment TURBT.

### 2.2. Variables and Outcome of Interest

The following variables of interest were recorded for all patients: age at diagnosis (years), sex, tumor size, tumor grade, and treatment type. The primary endpoint of the current study consisted of addressing CSM differences according to the treatment modality. The cancer-specific mortality definition relied on the SEER cause of death code (death due to bladder cancer).

### 2.3. Statistical Analyses

The estimated annual percentage changes (EAPC) described temporal trends in trimodal therapy vs. TURBT + CT vs. TURBT + RT. Kaplan–Meier plots depicted CSM rates according to each treatment modality. Multivariable Cox regression models were utilized to analyze CSM, with covariates including age at diagnosis, tumor size, tumor grade, and treatment modality. The same methodology was used in a subgroup analysis of TURBT + CT only vs. TURBT + RT only patients. All tests were two-sided, with a significance level of *p* < 0.05. R software environment for statistical computing and graphics (R version 4.2.2, R Foundation for Statical Computing, Vienna, Austria) was used for all analyses [21].

## 3. Results

### 3.1. Descriptive Characteristics of the Study Population

Overall, we identified 9358 patients with organ-confined urothelial carcinoma of the urinary bladder between 2004 and 2020 (Table 1). Of these, 3729 (40%) were treated with trimodal therapy vs. 4030 (43%) with TURBT + CT vs. 1599 (17%) with TURBT + RT. Relative to trimodal therapy, TURBT + CT patients were younger (median age 71 vs. 77 years), whereas TURBT + RT patients were older (median age 77 vs. 82 years, *p* < 0.001). Median tumor size was the same in trimodal therapy and TURBT + CT patients (4.0 cm), but larger in TURBT + RT patients (4.9 cm, *p* < 0.001).

### 3.2. Rate of Trimodal Therapy vs. TURBT + CT vs. TURBT + RT in Patients with Organ-Confined Urothelial Carcinoma of the Urinary Bladder 

Relative to the entire cohort of 9358 patients with organ-confined urothelial carcinoma of the urinary bladder, the proportion of trimodal therapy-treated patients increased over time from 37.2 to 48.4% (EAPC: 1.2%, *p* = 0.01, Figure 1). Similarly, the proportion of patients treated with TURBT + CT also increased from 33.1 to 41.3% (EAPC: 1.5%, *p* = 0.01). Conversely, the proportion of TURBT + RT patients decreased over time from 29.7 to 10.3% (EAPC: −6.0%, *p* < 0.001).

### 3.3. Survival Analyses Addressing CSM in Patients with Organ-Confined Urothelial Carcinoma of the Urinary Bladder According to Treatment Modality

Kaplan–Meier plots showed a median CSM of 60, 61, and 21 months in trimodal therapy, TURBT + CT, and TURBT + RT patients, respectively (Figure 2). In multivariable Cox regression models, relative to trimodal therapy, TURBT + CT independently predicted 1.14-fold higher CSM (95% Confidence Interval [CI]: 1.02, 1.25, *p* = 0.01, Table 2), after most complete adjustments for age at diagnosis, tumor size, and tumor grade. Similarly, TURBT + RT exhibited 1.68-fold higher CSM (95% CI: 1.49, 1.89, *p* < 0.001).

### 3.4. Survival Analyses Addressing CSM in Patients with Organ-Confined Urothelial Carcinoma of the Urinary Bladder According to Treatment Modality in the Subgroup of TURBT + CT vs. TURBT + RT

In multivariable Cox regression models in the subgroup analysis comparing TURBT + RT to TURBT + CT, TURBT + RT independently predicted 1.43-fold higher CSM (95% CI: 1.25, 1.63, *p* < 0.001) after most complete adjustments for age at diagnosis, tumor size, and tumor grade (Table 3).

## 4. Discussion

Recently, bladder-sparing strategies such as trimodal therapy have become more popular as they can achieve favorable cancer control outcomes with the preservation of organ function and quality of life [9,22]. However, scarce information exists about the use of CT or RT in isolation after TURBT, instead of CT and RT combination. Moreover, it is unknown to what extent cancer control rates after TURBT with exclusive CT or TURBT with exclusive RT may differ relative to strict trimodal therapy that includes both CT and RT after TURBT. We addressed these knowledge gaps and made several noteworthy observations.

First, within the current cohort of 9358 patients with organ-confined urothelial carcinoma of the urinary bladder, 3729 (40%) were treated with trimodal therapy, 4030 (43%) with TURBT + CT, and 1599 (17%) with TURBT + RT. These rates were surprising, as trimodal therapy did not emerge as the most frequently used treatment modality. Indeed, several previous population-based repositories addressed trimodal therapy as the principal non-surgical bladder-sparing multimodal treatment modality for patients with organ-confined urothelial carcinoma of the urinary bladder [23,24,25]. Instead, TURBT + CT was used most frequently. This elevated rate of exclusive CT use after TURBT is unexpected. For example, Deuker et al. evaluated RC vs. multimodal treatment modalities in T2N0M0 patients relying on the SEER database (2004–2016) [24]. In their study, the authors recorded that TMT was the most frequently used treatment modality after the standard of care (namely, RC). However, among all the non-surgical treatment modalities included, the authors did not take into consideration the combination of TURBT plus CT. In consequence, a direct comparison cannot be made. Unfortunately, this predominant use of CT alone after TURBT relative to CT combined with RT cannot be directly compared to any previous reports relying on other large-scale databases since they do not exist. Conversely, the much lower TURBT + RT rate is consistent with expectations [24]. Indeed, Deuker et al. recorded that RT after TURBT without CT was the least prevalent among all bladder-sparing multimodal treatments. However, the proportion of RT alone after TURBT recorded within the current study cannot be compared with reports from sources other than the SEER database.

Second, we also recorded patients’ characteristics and differences between the three examined subgroups of the current study cohort. Specifically, relative to trimodal-therapy patients, TURBT + CT patients were significantly younger (median age 77 years vs. 71 years), whereas TURBT + RT patients were older (median age 77 vs. 82 years). This observation is also consistent with previous population-based studies from SEER and NCDB, where the median age in TURBT + CT and TMT ranged from 70 to 77, while RT patients were much older [23,24,26,27,28].

Third, we also addressed trimodal therapy use vs. TURBT + CT vs. TURBT + RT over time. Specifically, we recorded a significant increase in trimodal therapy use over time from 37.2 to 48.4% (EAPC: +1.2%, *p* = 0.01). Conversely, TURBT + RT decreased over time from 29.7 to 10.3% (EAPC: −6.0%, *p* < 0.001). These observations are consistent with current clinical practice, where in recent years, trimodal therapy has represented the most used non-surgical bladder-sparing multimodal treatment. Moreover, these observations are also consistent with previous population-based analyses relying on the SEER database, where trimodal therapy use significantly increased over time in patients with organ-confined urothelial carcinoma of the urinary bladder [24] Conversely, TURBT + RT decreased over time from 29.7 to 10.3% (EAPC: −6.0%, *p* < 0.001). This observation is also consistent with other previous population-based analyses that recorded decreased TURBT + RT use in patients with organ-confined urothelial carcinoma of the urinary bladder. Interestingly, the steepest inflection in TURBT + RT use was recorded after the publication of the BC2001 trial [13]. Consistently, trimodal therapy use heavily increased during the same time span. Finally, within the current study, we recorded that TURBT + CT utilization increased from 33.1 to 41.3% (EAPC: 1.5%, *p* = 0.01). To the best of our knowledge, no previous studies have evaluated TURBT + CT use over time in patients with organ-confined urothelial carcinoma of the urinary bladder. In consequence, a direct comparison cannot be made. Additionally, any direct comparisons cannot be made with other analyses from data sources other than SEER. Ideally, other large-scale epidemiological databases such as NCDB should be used to validate the recorded observations.

Fourth, we tested for CSM differences in patients with organ-confined urothelial carcinoma of the urinary bladder according to trimodal therapy vs. TURBT + CT vs. TURBT + RT. Specifically, relative to trimodal therapy, TURBT + CT showed virtually the same CSM: median survival of 60 vs. 61 months, respectively. Conversely, TURBT + RT patients exhibited significantly higher CSM: median survival of 21 months. However, after multivariable adjustment for age, tumor size, and tumor grade, both TURBT + CT and TURBT + RT independently predicted higher CSM compared to trimodal therapy (HR: 1.14 and 1.68, respectively). Thus, it could be postulated that worse survival may be achieved when patients undergo CT alone after TURBT or RT alone after TURBT.

Subsequently, we tested for CSM differences in a subgroup analysis comparing TURBT + RT to TURBT + CT. Specifically, after multivariable adjustment for age, tumor size, and tumor grade, TURBT + RT independently predicted 1.43-fold higher CSM relative to TURBT + CT. To the best of our knowledge, no previous studies have investigated CSM differences in patients with organ-confined urothelial carcinoma of the urinary bladder treated with TURBT plus RT alone vs. TURBT plus CT alone using CSM as an endpoint. In consequence, a direct comparison with other previous publications cannot be made since they do not exist.

Taken together, these observations indicate that optimal cancer-control outcomes can be reached in patients with organ-confined urothelial carcinoma of the urinary bladder only when the most complete combination of non-surgical treatment modalities is offered: namely, TURBT followed by CT and RT combination. In consequence, trimodal therapy that relies on these three management components should be ideally recommended to all individuals in whom radical cystectomy is either not applicable or cannot be considered. However, in some select patients, either the RT component or CT cannot be given. Unfortunately, clinical decision making and/or patient comorbidities and performance status that could have influenced the treatment choice could not be accounted for in the SEER database. Under such premises and considering the data in our possession, it could be postulated that the combination of TURBT plus CT offers more favorable survival than TURBT plus RT.

To the best of our knowledge, no previous studies have specifically quantified the differences between trimodal therapies that rely on all three management elements components (namely TURBT, CT, and RT). Consequently, a direct comparison cannot be made. Indeed, no study relied on these three management components relative to CT or RT administered in isolation after TURBT. Thus, the magnitude of survival disadvantage relative to trimodal therapy cannot be compared with existing data. This consideration underlines the novelty of the current analysis.

The results of the current study should be considered in light of existing data. These data suggest that RT should ideally not be administered alone but should be accompanied by CT [13,14,29,30]. For example, despite the evident limitation comparing the current manuscript with randomized trials, our results are consistent with the evidence provided by the BC2001 trial [13]. Specifically, the five-year survival outcomes from BC2001 (49% in TMT vs. 37% TURBT + RT) are somewhat similar to the 5-year survival outcomes from our analysis (50% in TMT vs. 30% in TURBT + RT). However, relative to the BC2001 trial, it was not possible to account for patient comorbidities and other relevant clinical factors as could be inferred from the early divergence of the curves in our analysis. In a more recent report evaluating 10-year oncologic outcomes in the BC2001 trial, Hall et al. showed remarkable cancer-control benefits associated with the addition of CT to RT, such as improved locoregional control (HR: 0.6), lower disease recurrence (HR:0.78), higher cystectomy-free survival (HR:0.54), and lower CSM, although these were not statistically significant (HR: 0.79, *p* = 0.11) [14]. These observations suggest a pivotal role for the inclusion of CT in trimodal therapy regiments. Moreover, the pivotal role of CT in trimodal therapy regiments is also consistent with recent large population-based repositories such as SEER or the National Cancer Database (NCDB), where the addition of CT to RT resulted in lower CSM and lower overall mortality [24,31]. For example, Korpics et al. evaluated the survival benefit of concurrent chemotherapy in elderly patients with UCUB undergoing radiotherapy from NCDB. In their study, the authors showed a statistically significant and clinically meaningful survival benefit in patients receiving concomitant CT relative to RT alone (HR:0.74, *p* < 0.001) [31]. Similar considerations were made by Deuker et al., where RT without CT resulted in dismal survival relative to RC and TMT [24]. It is of note that a decrease in overall mortality after the addition of CT to TMT regiments could only be documented in population-based data repositories that offer greater maturity than randomized trials.

Conversely, the results of the current study also indicate that a relatively small disadvantage is associated with the exclusive use of CT after TURBT instead of TURBT followed by CT and RT. Unfortunately, other studies that examined cancer-control rates when exclusive CT after TURBT is applied are extremely scarce [28,32,33,34,35]. Most of these studies focused on evaluating cancer-control outcomes in a select population of patients who were complete responders to neoadjuvant chemotherapy (namely, pT0). In this select cohort of patients, five-year survival rates ranged between 69 and 90% [32,33,34]. Herr et al. examined this modality in a small, single-arm study of 63 individuals who refused RC. Here, TURBT was followed by cisplatin-based chemotherapy, with 36% five-year CSM rates [35]. Additionally, Audenet et al. performed a large-scale NCDB-based analysis of CT after TURBT [28]. Unfortunately, the authors did not provide CSM rates but rather overall survival. Furthermore, they also did not compare cancer-control outcomes to other non-surgical treatments as was performed in the current study. Consequently, the current data regarding CT alone after TURBT represent a first and should ideally be validated in future analyses.

Despite the novelty of our findings, several limitations need to be acknowledged. First, it is retrospective in nature. However, this limitation is shared with most similar previous reports addressing the oncologic control of trimodal therapy, which are retrospective in nature. Indeed, in the absence of prospective comparative trials comparing bladder-sparing strategies, only large-scale retrospective studies could provide valuable clinical evidence among these therapies. The second limitation consists of the lack of detail about the specific staging of the primary tumor. Indeed, within the SEER database, it is not possible to assess tumor depth of invasion, eventual multifocality of the tumor, and eventual complete resection after TURBT. Unfortunately, we cannot account for this information as it is not available in the SEER database. Third, the SEER database does not provide information about baseline comorbidities. Consequently, more detailed analyses where comorbidities could be applied were not possible, thus possibly affecting survival outcomes in older and more frail patients. Additionally, the type, dose, and timing of radiotherapy, as well as chemotherapy, were not available. In consequence, the current observations are only applicable to population-based analysis. Nonetheless, they provide valuable and encouraging insight into trimodal therapy outcomes. Similarly, information regarding the eventual administration of immune checkpoint inhibitors (ICIs) is also not available in the SEER database. However, current guidelines do not recommend ICI administration in T2N0M0 patients. Indeed, since their approval in 2016, ICIs have been administered in these patients only in the context of prospective trials [36]. Conversely, within the current study, we included patients from 2004 to 2020. Therefore, taking the above considerations into account, it is unlikely that a remarkable proportion of patients identified in the current study were treated with ICIs. Moreover, the SEER database does not provide information on the main purpose of non-surgical treatments (curative versus palliative purposes). Last but not least, the characteristics of the SEER population reflect urothelial carcinoma patients from the United States that may not be comparable to urothelial carcinoma patients from other geopolitical regions. It is also possible that SEER patients do not reflect the characteristics and treatment outcomes of patients treated in centers of excellence within the United States.

## 5. Conclusions

When radical cystectomy is not an option, strict trimodal therapy that includes both CT and RT after TURBT offers the best cancer control. When strict trimodal therapy cannot be delivered, cancer-specific survival outcomes appear to be superior with TURBT + chemotherapy compared to TURBT + RT.

## Figures and Tables

**Figure 1 cancers-16-01292-f001:**
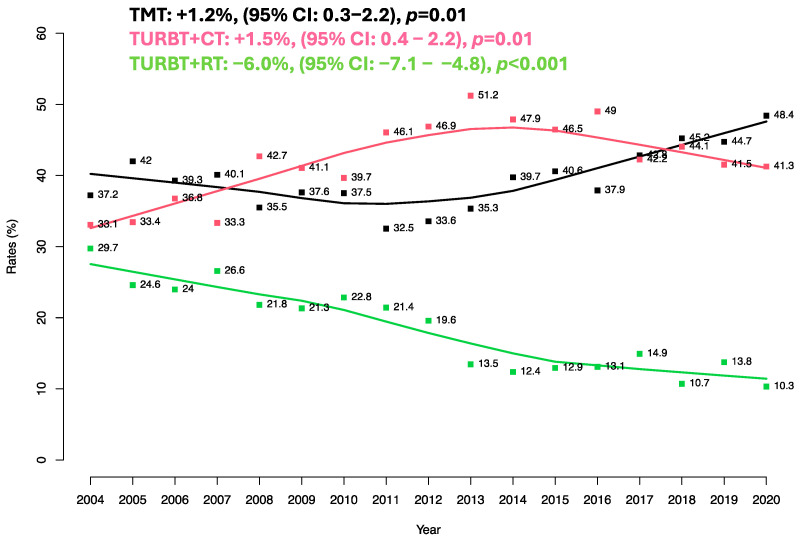
Estimated annual percent changes (EAPC) depicting temporal trends in patients with organ-confined urothelial carcinoma of the urinary bladder treated with trimodal therapy (TMT) vs. chemotherapy after TURBT (CT) vs. radiotherapy after TURBT (RT).

**Figure 2 cancers-16-01292-f002:**
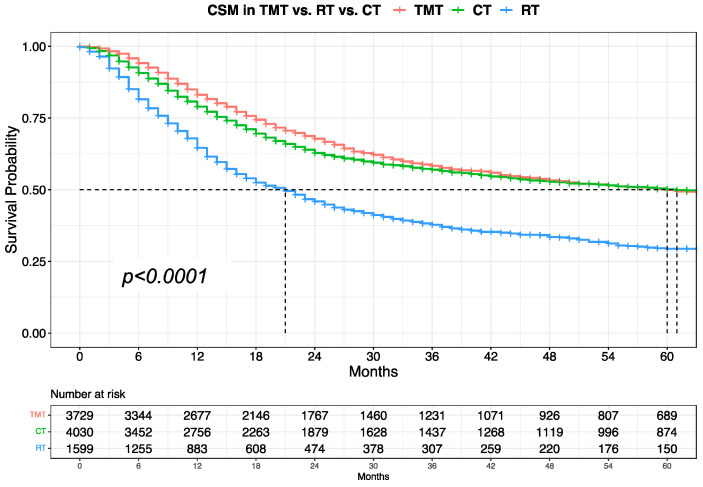
Kaplan–Meier analyses depicting CSM estimates in patients with organ-confined urothelial carcinoma of the urinary bladder treated with trimodal therapy (TMT) vs. chemotherapy after TURBT (CT) vs. radiotherapy after TURBT (RT).

**Table 1 cancers-16-01292-t001:** Baseline characteristics of 9358 patients with organ-confined urothelial carcinoma of the urinary bladder treated with trimodal therapy (TMT) vs. chemotherapy after TURBT (TURBT + CT) vs. radiotherapy after TURBT (TURBT + RT) from the Surveillance, Epidemiology, and End Results database (2004–2020).

Characteristic	TMT 3729 (40%) ^1^	TURBT + CT 4030 (43%) ^1^	TURBT + RT 1599 (17%) ^1^	*p*-Value ^2^
Age	77 (70, 82)	71 (63, 78)	82 (76, 86)	<0.001
Size	4 (30, 51)	4 (27, 51)	4.9 (30, 54)	<0.001
Sex				0.002
Male	2789 (75%)	3079 (76%)	1150 (72%)	
Female	940 (25%)	951 (24%)	449 (28%)	
Grade				0.002
High grade	3608 (98%)	3821 (97%)	1509 (97%)	
Low grade	67 (2%)	120 (3%)	43 (3%)	

^1^ Median (IQR); n (%). ^2^ Kruskal-Wallis rank sum test; Pearson’s Chi-square test.

**Table 2 cancers-16-01292-t002:** Multivariable Cox regression models testing predictors of cancer-specific mortality in OC UCUB patients according to treatment modality after adjustment for age at diagnosis, tumor size, and tumor grade.

Characteristic	HR ^1^	95% CI ^1^	*p*-Value
Treatment			
TMT	-	-	-
CT	1.137	1.029, 1.256	0.012
RT	1.682	1.491, 1.898	<0.001

^1^ HR = Hazard Ratio, CI = Confidence Interval.

**Table 3 cancers-16-01292-t003:** Multivariable Cox regression models testing predictors of cancer-specific mortality in OC UCUB patients treated with chemotherapy after TURBT (CT) vs. radiotherapy after TURBT (RT) after adjustment for age, tumor size, and tumor grade.

Characteristic	HR ^1^	95% CI ^1^	*p*-Value
Treatment modality			
CT	-	-	-
RT	1.428	1.249, 1.632	<0.001

^1^ HR = Hazard Ratio, CI = Confidence Interval.

## Data Availability

The data presented in this study are available in this article.

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
