# Peer review of "Non-Surgical Bladder-Sparing Multimodal Management in Organ-Confined Urothelial Carcinoma of the Urinary Bladder: A Population-Based Analysis"

_cancers, 2024, doi:10.3390/cancers16071292_

Round 1

Reviewer 1 Report

Comments and Suggestions for Authors

A very well presented manuscript on the role of trimodality treatment, TURBT+Chemotherapy and TURBT+Radiotherapy alone for patients with T2N0M0 disease who did not undergo Radical cystoprostatectomy. Authors very clearly presented a methodologically sound and interesting manuscript, therefore should be commended on that.

The research addresses clearly the main question. Most research in bladder preservation treatment compares radical cystectomy with complete trimodal treatment, while the present study also compares parts of trimodal study without the whole scheme of surgery-chemotherapy-radiotherapy, with an impressive sample size coming from a registry.

As stated above, this study compares not only trimodal therapy as a whole but also parts of this regimen for patients who cannot receive chemotherapy or radiotherapy, resembling real life scenarios and that is another strong point of the study.

I think that study is very well designed with limitations already acknowledged by authors and I don't find anything that should be added. Conclusions are consistent with the findings of authors with complete trimodal regimen being superior to surgery+chemotherapy and this being superior to surgery+radiotherapy. This is not well debated in current literature and this study provides a very clear insight based on a large sample size. Tables and figures are very well designed.

In general I believe this is a very valid study and well-written manuscript.

Author Response

We sincerely thank the Reviewer for these nice comments and the positive feedback on our work 

Reviewer 2 Report

Comments and Suggestions for Authors The authors present an interesting population based analysis based on SEER data of comparative survival outcomes of non-cystectomy patients with T2N0 MIBC treated with either trimodality therapy with chemoradiation vs chemo alone or RT alone after TURBT. Comments below to consider. There are a number of grammatical errors throughout the manuscript, some corrections below but suggest additional proofreading.    line 46: do you mean non-extirpative? line 49: treated with line 53: need to define EAPC line 59: offers the line 84: perhaps preferred is a better term than indicated. For practical reasons, considering age/comorbidities/performance status, we unfortunately do manage some patients with more conservative unimodality treatment (such as palliative TURBTs alone).  line 99: re: "the adjacent lymph nodes." It remains controversial whether there is a benefit to elective pelvic nodal RT in the setting of bladder preservation chemoradiation. Patterns of practice vary significantly. Many of the landmark trials including BC2001 did not include pelvic nodal radiation and treated bladder alone. May be more appropriate to state without or without the adjacent lymph nodes.  line 100-101: Concurrent chemo with RT in this setting primarily has demonstrated improvement in LRC, so the data is more in support of benefit in terms of radiosensitization. In BC2001, even at 10 year update, there was not statistically significant benefit in terms of MFS with addition of chemo. May be worthwhile to acknowledge this point or frame the point that neoadjuvant chemotherapy regimens prior to cystectomy do show this benefit, but how to integrate or best sequence these approaches in the setting of TMT is still a subject of investigation.  line 105: in patients who are candidates line 109: none have quantified line 128: remove "of" line 154-155: would mention that the age differences between these cohorts was statistically significantly different  line 155: shouldn't this be (median age 82 vs 77 years) Tabie1: size should have a units label. Would also consider using cm, it is not as common to use measurements of mm.  line 157: would mention that this was statistically significantly different (larger for TURBT+RT) line 188: How is it that the p value is significant for CSM difference for TURBT + CT vs TMT when the curves are essentially superimposed from year 3-5? Figure 2: why is the p value in the figure discordant from then p value in the table? table 3: appears identical to table 2? line 241: these line 251: relying on line 259: least prevalent line 263: subgroups of the  line 281: This is not exactly accurate, would rephrase. The BC2001 trial numerically did show higher OS with combined modality therapy vs RT alone, however, this was not statistically significant p=0.16. Even at 10 years follow up, there is still no statistically significant improvement. However, it is possible that the study was underpowered for this and the difference is real.  Agree it is worth pointing out that the inflection point of growth for TMT does appear to correspond exactly to the publication year of BC2001 in 2012.  lines 292-293: These statements appear contradictory, suggest rephrase. "These comparisons revealed higher CSM in TURBT+CT and TURBT+RT. Specifically, relative to trimodal therapy, TURBT+CT showed virtually the same CSM: median survival 293 60 versus 61 months, respectively." lines 298-299: Would state less strongly, recognizing the limitations of this retrospective SEER analysis and likelihood of significant confounders and patient selection biases.  lines 313-314: suggest authors discuss some of these practical considerations of who comprises these cohorts of patients not receiving traditional TMT that are receiving CT alone or RT alone. This is an important underpinning of the study and may significantly influence the survival differences suggested in this analysis.  lines 315-316: speculative, still likelihood of significant confounders and patient selection biases, need to acknowledge.  lines 323-324: Recognizing limitations in a direct comparison, it may be useful to compare with actually with the BC2001 which is the only randomized trial to compare RT +/- chemo. I would point out that it appears that the 5 year survival outcomes from BC2001 (RT+chemo 49% vs 37%) are somewhat similar to the 5 yr outcomes from your analysis (roughly appears to be about 50% vs 30%). Interestingly, in BC2001, the survival curves do not begin to separate until about 2-3 years. However, in this SEER analysis, it appears that the curves almost immediately separate, already apparent at 6 mos. This would be unexpected to have CSM differences attributable to simply treatment pathway strategy for localized T2N0 disease. It suggests that there are inherent differences in disease and patient background to begin with accounting for the poorer outcomes in the TURBT + RT cohort of pts.  line 326: Would cite the actual trial name, BC2001 given that is a landmark trial and only one of this design. Also, misleading wording here, this is not a "recent trial". This is long-term follow up of the study originally reported in 2012.  line 329: the CSM difference was a trend but not statistically significant in that trial, p=0.11, should acknowledge this.  line 333: please expand on these studies you have quoted and how they support your statement here line 335-336: interesting statement, explain your rationale. Agree that randomized trials are challenging to conduct in this space to address this question. -line 343: confusing wording, neither do not, need to rephrase -line 340: There are some scattered other publications relevant to this topic. Consider these:   Cancer. 2003 Apr 1;97(7):1644-52. doi: 10.1002/cncr.11232.   Urology. 2018 Jan:111:116-121. doi: 10.1016/j.urology.2017.09.003. Epub 2017 Oct 12.   J Urol. 2018 Nov;200(5):1005-1013. doi: 10.1016/j.juro.2018.05.078. Epub 2018 May 19.   -looking forward, this may be an opportunity here as well to briefly mention some of the new data in this space of systemic therapy alone for bladder preservation including trials evaluating chemotherapy + immunotherapy combinations, such as  https://www.nature.com/articles/s41591-023-02568-1   line 349-350: would qualify this statement a bit, there is a large body of literature of prospective data reporting on oncologic outcomes of trimodality therapy. What is lacking are prospective comparative trials of strategies and randomized trials.  line 363: characteristics and treatment outcomes  line 368-369: I would suggest tempering some of the strength of these conclusions. There are clearly significant limitations of a population based retrospective study like this and any SEER analysis, and the findings are often are not consistent with what is ultimately demonstrated from level 1 evidence from prospective studies or randomized trials. Suggest stating something like: The findings from this SEER population based review suggest that when traditional trimodality therapy cannot be delivered, cancer specific survival outcomes appear to be superior with TURBT + chemotherapy compared to TURBT + RT.    -In practice, considering the experience at our own center, we manage most patients with localized T2 MIBC with neoadjuvant chemotherapy followed by cystectomy. In those managed nonsurgically, we treat select patients with TMT. The remainder of the patients are more commonly treated with TURBT + RT alone, typically older, more frail patients that are not candidates for concurrent systemic therapy. At our institution, we actually manage very few patients with TURBT followed by chemotherapy alone, so I am surprised that the proportion of patients in this cohort is as large as it is. We do have patients that had neoadjuvant chemotherapy, then refused surgery. This may also include some patients that undergo palliative systemic therapy alone (but this is more commonly immunotherapy in those with more locally advanced disease). Can you comment on what patients you feel comprise this cohort of patients undergoing TURBT + CT, and what would make them ineligible for or not to have received RT. In our practice, we manage a number of non-cystectomy candidate patients with TMT. Often, even in older patients with significant comorbidities, RT + concurrent gemcitabine or RT + 5FU/MMC may be well-tolerated. Those that we manage with RT alone are typically those that are very poor performance status and much more medically frail and with more advanced disease. It is expected that this cohort of patients would have more adverse outcomes. This seems in keeping with the fact that this SEER cohort of TURBT consisted of pts that were older and with larger tumors. I think this point should be discussed a bit further as it likely is a factor influencing the outcomes here.  Comments on the Quality of English Language

As above, a number of grammatical corrections needed and additional proofreading

Author Response

We sincerely express our gratitude to the Reviewer’s effort and careful contribution. The above considerations and suggestions are extremely helpful in improving the quality of our work and strengthening the value of our findings. We modified pur manuscript accordingly. 

 However, due to the numerous suggestions received, we have limited our responses to addressing only those comments that required further discussion. We hope that our answers address all concerns raised by the reviewer.

line 188: How is it that the p value is significant for CSM difference for TURBT + CT vs TMT when the curves are essentially superimposed from year 3-5?
Long-rank test showed in figure 2 is testing any difference between the three groups. Therefore, the significance of p-value is driven by the difference recorded in the third group (TURBT+RT)

Figure 2: why is the p value in the figure discordant from then p value in the table?
The KM curve reflect a univariable stratification of the observed CSM over time, while the table displays a multivariable model testing for the association between CSM and treatment modalities accounting for many observed demographic and clinical variables that could confound our outcome. While this two should offer similar interpretation, the statistical tests are different. Therefore, the p-value will not be the same.

table 3: appears identical to table 2?
We thank the Reviewer for noticing this error.  Unfortunately, there was an error during the finalization phase of the manuscrpit, resulting in the same table being unintentionally uploaded twice. We sincerely apologize to the reviewer for any oversights or shortcomings in our manuscript. The correct Table has been added accordingly.  

lines 292-293: These statements appear contradictory
We have eliminated the sentence "These comparisons revealed higher CSM in TURBT+CT and TURBT+R." to make it clearer

335-336: interesting statement, explain your rationale. Agree that randomized trials are challenging to conduct in this space to address this question.

Regarding this statement, two fundamental differences should be acknowledged when comparing population-based analyses and randomized trials addressing survival benefit after TMT. It is of note that randomized trials provide the best evidence and, ideally, relevant clinical question such as the survival benefit after TMT relative to other treatment modalities should only rely on such evidence. However, randomized trials in this context are not always possible (i.e. SPARE trial in TMT versus RC). Moreover, clinical trials often do not reflect the real-life scenario about current clinical practice, which instead can be provided by population-based analyses. On the other hand, of course there are several clinical information and confounder we cannot account for when relying on population-based analyses. In consequence, all observations provided should be taken cum grano salis

 -looking forward, this may be an opportunity here as well to briefly mention some of the new data in this space of systemic therapy alone for bladder preservation including trials evaluating chemotherapy + immunotherapy combinations, such as  https://www.nature.com/articles/s41591-023-02568-1

We thank the Reviewer for the valuable suggestion. However, in this case we think that discussing the latest evidence regarding the association of chemotherapy and immunotherapy exceeds from the intention of our study, which is only to compare the survival differences between trimodal therapy and other more established treatments modalities. Our concerns is that discussing the value of chemotherapy+immunotherapy administration in this context could results somehow confusing for the readers. Moreover, it could also expose our study to criticism, as immunotherapy is a variable we cannot account for in the SEER database. 

At our institution, we actually manage very few patients with TURBT followed by chemotherapy alone, so I am surprised that the proportion of patients in this cohort is as large as it is. We do have patients that had neoadjuvant chemotherapy, then refused surgery. 

We totally agree with Reviewer's point of view. We also were surprised about the large sample size of these patients. It could be postulated that a proportion of these patients could have received neoadjuvant chemotherapy and then refused to undergo RC. Unfortunately, clinical decision making or any other informations that could explain the reason behind the large number of this cohort are not provided in the SEER database. However, although more informations are not available, it still reflect real-life scenarios, where a non-negligible proportion of patients received TURBT+CT as definitive treatment, regardless of the reason behind it. As such, we believe it is still worth it to report it, although we cannot provide a formal and more precise informations to explain it.

 This may also include some patients that undergo palliative systemic therapy alone

We partially agree with reviewer's point of view. However, we do not think that these patients underwent palliative systemic therapy alone. First, because there is a low probability that T2N0M0 patients require palliative systemic therapy within five-years after initial diagnosis. Second, within the current study, five-year cancer-specific survival rates  in TURBT+CT patients are not that different relative to TMT-treated patients. These observations suggest that most of TURBT+CT patients did not significantly differ relative to TMT counterparts. However, since we do not have specific informations about systemic therapy timing and setting, we could not totally exclude this explaination, although it is not the most likely option. 

Those that we manage with RT alone are typically those that are very poor performance status and much more medically frail and with more advanced disease. It is expected that this cohort of patients would have more adverse outcomes. This seems in keeping with the fact that this SEER cohort of TURBT consisted of pts that were older and with larger tumors. I think this point should be discussed a bit further as it likely is a factor influencing the outcomes here. 

We totally agree with Reviewer's point of view. We further discussed this aspect in the limitation section accordingly